# Quantitative Analysis of Polysaccharide Composition in *Polyporus umbellatus* by HPLC–ESI–TOF–MS

**DOI:** 10.3390/molecules24142526

**Published:** 2019-07-10

**Authors:** Ning Guo, Zongli Bai, Weijuan Jia, Jianhua Sun, Wanwan Wang, Shizhong Chen, Hong Wang

**Affiliations:** 1School of Pharmaceutical Sciences, Peking University, Beijing 100191, China; 2Kangmei Pharmaceutical Co.Ltd, Puning 515300, China

**Keywords:** composition analysis, HPLC–MS, monosaccharide hydrolysis, *Polyporus umbellatus*, polysaccharide

## Abstract

*Polyporus umbellatus* is a well-known and important medicinal fungus in Asia. Its polysaccharides possess interesting bioactivities such as antitumor, antioxidant, hepatoprotective and immunomodulatory effects. A qualitative and quantitative method has been established for the analysis of 12 monosaccharides comprising polysaccharides of *Polyporus umbellatus* based on high-performance liquid chromatography coupled with electrospray ionization–ion trap–time of flight–mass spectrometry. The hydrolysis conditions of the polysaccharides were optimized by orthogonal design. The results of optimized hydrolysis were as follows: neutral sugars and uronic acids 4 mol/L trifluoroacetic acid (TFA), 6 h, 120 °C; and amino sugars 3 mol/L TFA, 3 h, 100 °C. The resulting monosaccharides derivatized with 1-phenyl-3-methyl-5-pyrazolone have been well separated and analyzed by the established method. Identification of the monosaccharides was carried out by analyzing the mass spectral behaviors and chromatography characteristics of 1-phenyl-3-methyl-5-pyrazolone labeled monosaccharides. The results showed that polysaccharides in *Polyporus umbellatus* were composed of mannose, glucosamine, rhamnose, ribose, lyxose, erythrose, glucuronic acid, galacturonic acid, glucose, galactose, xylose, and fucose. Quantitative recoveries of these monosaccharides in the samples were in the range of 96.10–103.70%. This method is simple, accurate, and sensitive for the identification and quantification of monosaccharides, and can be applied to the quality control of *Polyporusumbellatus* as a natural medicine.

## 1. Introduction

*Polyporus umbellatus* (Pers.) Fries, also known as “Zhu Ling” in China, belongs to the *Polyporaceae* family of the Basidiomycota phylum [1,2]. It is a widely used medicinal fungus in Asia, especially in China and Japan, and has been one of the most important traditional Chinese medicines for more than 2500 years [3]. The sclerotia is the main medicinal parts of *Polyporus umbellatus*, and has been used as a diuretic for the treatment of diseases such as kidney, edema, scant urine, and vaginal discharge, as well as diarrhea and jaundice [4]. Numerous pharmacological effects including anticancer, immuno-enhancing, hepatoprotective, radio-protective, and antioxidative activities have been reported [5,6,7,8,9,10]. To our knowledge, the physiological activities of polysaccharides depend on the composition of monosaccharides [11], and identifying this composition is the very first step to unveil relevant physicochemical properties, and structure–activity relationships. In addition, monosaccharide composition is an essential parameter for the quality evaluation of polysaccharide [12].

A great number of analytical techniques have been used to detect monosaccharides in complex samples, including thin-layer chromatography (TLC) [13], gas chromatography (GC) [14], high-performance liquid chromatography (HPLC) [15,16,17,18,19], capillary electrophoresis (CE) [20,21,22], gas chromatography–mass spectrometry(GC–MS) [23,24], and liquid chromatography–mass spectrometry (LC–MS) [25,26,27]. HPLC has been accepted as one of the common techniques for the analysis of sugars, which was used for simultaneous determination of neutral, acidic, and basic carbohydrates [28,29]. However, the lack of chromophores or fluorophores in the structure of monosaccharides prevents direct detection by UV or fluorescence detectors. Refractive index detection and other related methods do not often meet the demands of modern analysis with regard to sensitivity and/or selectivity [30,31]. Therefore, the derivatization of monosaccharides is indispensable to obtain highly sensitive detection [32,33]. The reagent 1-phenyl-3-methyl-5-pyrazolone (PMP), first developed in 1989 by Honda’s group [32], is a popular label that reacts with reducing carbohydrate under mild conditions, requiring no acid catalyst and causing no desialylation and isomerization [32,34,35,36]. In addition, pre-column derivatization with the PMP method was first developed for the analysis of carbohydrates by HPLC [37]. The derivatized monosaccharides exhibit strong ultraviolet (UV) absorption at 245 nm and higher hydrophobicity than their native form, especially compatible with HPLC separation and UV detection. In the case of complex sugar composition, the HPLC–MS method with PMP derivatization can be applied to improve the detection sensitivity and specificity.

There are many studies on polysaccharides from *Polyporus umbellatus* [38,39,40,41]. However, the method of HPLC–MS with PMP precolumn derivatization has not been applied to *Polyporus umbellatus*. In this study, we firstly designed an orthogonal test with three factors and three levels for the optimization of the polysaccharide hydrolysis, then established a precolumn PMP derivatization HPLC–DAD–ESI–IT–TOF–MS method for the simultaneous determination of 12 sugars. In addition, the characteristic fragment ions of the 12 PMP-labeled monosaccharides were collected.

## 2. Results and Discussion

### 2.1. Identification of Monosaccharides Comprising Polysaccharide

#### 2.1.1. HPLC Separation

The HPLC separation conditions were adjusted and tested for a mixture of 12 PMP-labeled monosaccharides (mannose, glucosamine, lyxose, rhamnose, ribose, erythrose, glucuronic acid, galacturonic acid, glucose, galactose, xylose, and fucose). The elution system was chosen as 0.1 mol/L aqueous ammonium acetate (solvent B) (pH 4.5) and acetonitrile (solvent A), since no nonvolatile inorganic salts are allowed into the MS detector. In this study, the pH of ammoniumacetate solution and the percentage of acetonitrile were investigated to improve the separation. As a result, 12 present PMP-labeled monosaccharides were successfully separated in the order of mannose, glucosamine, rhamnose, ribose, lyxose, erythrose, glucuronic acid, galacturonic acid, glucose, galactose, xylose, and fucose within 180 min in this study (Figure 1). The structures of identified monosaccharides are displayed in Figure 2.

#### 2.1.2. ESI–IT–TOF–MS analysis

All PMP-labeled monosaccharides were characterized by ESI–IT–TOF–MS method (positive-ion mode, Figure 1). The numerical data are listed in Table 1, which suggests the fragmentation mode for PMP-labeled monosaccharides. In the current study, 12 PMP-labeled compounds were unambiguously identified based on their chromatographic and MS fragment behaviors (Figure 1), and by comparisons with the reference standards. According to molecular weights, common monosaccharides could be divided into six groups, namely hexose, hexosamine, pentose, tetrose, methylpentose, and hexuronic acid, whose PMP derivatives gave quasimolecular ions at *m/z* 511, 510, 481, 451, 495 and 525, respectively.

The data summarized in Table 1 along with the relative abundances of the peaks suggest the fragmentation pathways of PMP-labeled monosaccharides. All the observed fragments were subdivided into four groups. The first group comprises low abundance peaks which correspond to dehydration such as [M + H − H_2_O]^+^ (occurring in any monosaccharide). The second group comprises low abundance peaks corresponding to the loss of one PMP group, dehydration, and deamination such as in [M + H − PMP]^+^, [M + H – PMP − H_2_O]^+^, [M + H – PMP − 2H_2_O]^+^, and [M + H – PMP − 3H_2_O]^+^ (occurring in any monosaccharide), as well as [M + H − PMP − NH_3_]^+^, [M + H − PMP − NH_3_ − H_2_O]^+^ and [M + H – PMP − NH_3_ − 2H_2_O]^+^, typical for glucosamine. The third group comprises high abundance peaks which correspond to the loss of one PMP group and characteristic C-C bond cleavage of the carbohydrate skeleton. These are peaks at *m/z* 271 due to the loss of one PMP group and then the cleavage of the C5-C6 bond followed by loss of 2H_2_O or NH_3_ (typical for glucosamine), peaks at *m/z* 241 due to the loss of one PMP group and the cleavage of the C4-C5 bond followed by a loss of H_2_O or NH_3_ (typical for glucosamine), peaks at *m/z* 217 and 216 (only for glucosamine) due to the loss of one PMP group and the cleavage of the C2-C3 bond, and the strong peaks at *m/z* 187 due to the loss of one PMP group and the cleavage of the C1-C2 bond. The last group comprises high abundance peaks correspondent to base peak ions [PMP + H] ^+^ at *m/z* 175 and characteristic C-C bond cleavage of the carbohydrate skeleton. These are peaks at *m/z* 373 and 372 (only for glucosamine) due to the cleavage of the C2-C3 bond followed by loss of H_2_O, and the weak peaks at *m/z* 403 and 402 (only for glucosamine) due to the cleavage of the C3-C4 bond followed by loss of H_2_O (for more details, Table 1).

In the MS^2^ spectra, the fragments at *m/z* 372 (cleavage of C2-C3 in glucosamine) and *m/z* 402 (cleavage of C3-C4 in glucosamine) exhibit high abundance as an indication for the presence of glucosamine. Other types of monosaccharides can be identified by their fragmentation analysis.

### 2.2. Optimization of Hydrolysis Conditions

The PPS has a complex monosaccharide composition which may contain acidic, neutral, and basic sugars in one molecule. In the present study, the orthogonal L_9_ (3)^4^ experiments were designed to investigate optimal hydrolysis conditions of PPS. Glucosamine, glucuronic acid, and glucose were chosen on behalf of basic sugars, acidic sugars, and neutral sugars. The results of orthogonal test and extreme difference analysis are presented in Appendix A. The analysis of variance was performed by statistical software SPSS 20.0 and the results are listed in Appendix A. As for glucose, according to the R values, we found that hydrolysis time played the critical role in the hydrolysis process, followed by TFA concentration and hydrolysis temperature. Analysis of variance results (Appendix A) indicated that hydrolysis temperature, TFA concentration, and hydrolysis time had statistically significant effects on the hydrolysis process with 95% confidence, and the order of influence factors was consistent with the results of the intuitive analysis. As for glucosamine, the hydrolysis temperature was the major factor affecting the hydrolysis process, while the minor factors were the concentration of TFA and hydrolysis time. Analysis of variance results indicated that hydrolysis temperature had statistically significant effect on hydrolysis process with 95% confidence, while TFA concentration and hydrolysis time did not have significant effect. As for glucuronic acid, it was found that the effect of factors is C > A > B based on the R values. TFA concentration played the important role in the hydrolysis process, followed by hydrolysis temperature and hydrolysis time. Analysis of variance results indicated that three factors did not have significant effect on hydrolysis process. The optimized hydrolytic conditions of PPS determined by orthogonal experiment were as follows: neutral sugars: 4 mol/L TFA, 6 h, 120 °C; amino sugars: 3 mol/L TFA, 3 h, 100 °C; uronic acids: 4 mol/L TFA, 6 h, 120 °C.

### 2.3. Method Validation

The analytical method was validated in terms of linearity, detection limit, precision, stability, repeatability and recovery. A total of 10 standard monosaccharides (mannose, glucosamine, lyxose, erythrose, glucuronic acid, galacturonic acid, glucose, galactose, xylose, and fucose) were used for these tests.

#### 2.3.1. Calibration curves and limit of detection

All calibration curves were established by plotting the chromatographic peak area of monosaccharide derivatives versus the concentration of the corresponding monosaccharide solution shown in Table 2. As a consequence, the correlation coefficients (R^2^ >0.9991) indicate that all calibration curves had excellent linearities within the test ranges. Furthermore, the limit of detection (LOD) and limit of quantification (LOQ) of each analyte were determined as the concentration of standard solution with S/N = 3 (signal-to-noise ratio) and S/N = 10. The results showed that the LOD values of the 10 monosaccharides were in the range from 0.191 to 1.152 μmol/L (Table 2), indicating the sensitivity of the method.

#### 2.3.2. Precision, Reproducibility, and Stability

The precision was calculated as the relative standard deviation (RSD) for consecutively analyzing the same sample for six times. The repeatability was evaluated by six repeated determination of the hydrolyzed monosaccharides from polyporus polysaccharide. The stability was assessed by analyzing the same sample six times at different times (0, 3, 6, 9, 12, and 24 h). The results of precision, reproducibility, and stability are listed in Table 3. The RSD values for precision were 0.39–1.13%, which indicated that the method precision was satisfactory. The RSD values of the stability were less than 1.30%, indicating that the sample solution was stable within 24 h. The RSD of repeatability was within 0.49–1.39%.

#### 2.3.3. Recovery

Each monosaccharide solution was added to the hydrolyzed polysaccharide sample at a similar concentration of the same monosaccharide in the sample (*n* = 3). The recovery rate was calculated as follows: recovery (%) = [(amount detected- original amount)/amount spiked] × 100%. The RSD (%) was calculated according to the following equation: RSD (%) = (standard deviation/mean) × 100%. The average recoveries of all ten monosaccharides ranged from 96.10% to 103.70%, and the RSD values fell within 0.96–2.36% (Table 4). This accuracy is within the acceptable ranges. Results of the method validation described above indicate that the method is precise and accurate for the analysis of polysaccharide from *Polyporus umbellatus*.

### 2.4. Sample Analysis

To assess the quality of Polyporus umbellatus and obtain the composition of the polysaccharides, 12 samples of Polyporus umbellatus purchased from different natural habitats in China were analyzed in this study. The results of 12 samples are shown in Table 5.

All 12 batches of Polyporus umbellatus polysaccharide were found to be composed of mannose, glucosamine, rhamnose, ribose, lyxose, erythrose, glucuronic acid, galacturonic acid, glucose, galactose, xylose, and fucose. Compared to previous reports [5,39,40,41,42,43], the monosaccharide composition of PPS includes mannose, rhamnose, glucuronic acid, glucose, galactose, xylose, and fucose. However, glucosamine, ribose, lyxose, erythrose, and galacturonic acid have not been previously reported in PPS. In addition, the contents of monosaccharides were between 20.88 and 3329.49 nmol/mg. The predominant monosaccharides in PPS are glucose, galactose, xylose, fucose and erythrose. However, the contents of glucuronic acid and galacturonic acid are less than other monosaccharides. There are significant differences in the content and composition of the components across the samples. The reason for this phenomenon might be due to the differences in geographical origin and growth environment. Interestingly, the composition and order of monosaccharide contents were similar with each other. It was reported that terminal mannose, *N*-acetylglucosamine, or fucose in polysaccharides can be recognized by mannose receptors in macrophages, leading to elevated protective function of immune system [44,45,46,47,48]. This study showed that all PPS samples contain mannose and fucose, which may partially explain the immune-enhancement activity of the sample.

## 3. Materials and Methods

### 3.1. Chemicals and Reagents

*d*-Mannose, *d*-galactose, *d*-xylose, *l*-rhamnose, *d*-glucuronic acid, and *d*-glucosamine were purchased from the Harveybio Gene Technology Co., Ltd. (Beijing, China). *d*-galacturonic acid and *d*-fucose were obtained from Solarbio Technology (Beijing, China). *d*-Glucose, d-ribose, and d-lyxose were acquired from J&K scientific Ltd. (Beijing, China). d-Erythrose was from Shanghai Macklin Biochemical Co., Ltd. (Shanghai, China). The purities of all reference compounds except *D*-erythrose (75%) were found to be above 95% as determined by HPLC. 1-Phenyl-3-methyl-5-pyrazolone (PMP) and trifluoroacetic acid (TFA) were obtained from Shanghai Macklin Biochemical Co., Ltd. (Shanghai, China). MS grade acetonitrile (Merck, Darmstadt, Germany), MS grade formic acid (FA) (Fisher Scientific), and de-ionized water of 18.2 MΩ purified by a Milli-Q system (Milford, MA, USA) were used for the mobile phases.

A total of 12 batches of crude medicinal *Polyporus umbellatus* from different natural habitats in China were purchased, all of which were identified by Professor Hong Wang (Department of Natural Medicines, School of Pharmaceutical Sciences, Peking University) and the voucher specimens were deposited at the School of Pharmaceutical Sciences, Peking University. Their habitats and codes were as follows: Shaanxi (S1); Shanxi (S2); Shanxi (S3); Yunnan (S4); Yunnan (S5); Shaanxi (S6); Shaanxi (S7); Henan (S8); Hebei (S9); Sichuan (S10); Jilin (S11); Gansu (S12). Sample S1 was used for the method development.

### 3.2. Sample Preparation

The polysaccharide was extracted from *Polyporus umbellatus* by hot-water extraction and ethanol precipitation [49,50]. Briefly, the finely powder of dried sclerotia of *Polyporus umbellatus* (10 g) was defatted with 95% alcohol and dried (3 × 100 mL), then extracted with boiling water (2 × 300 mL) for 3 h. After filtration, the water extracts were combined and concentrated to 100 mL under reduced pressure. Then, 400 mL of ethanol were slowly added to the solution under stirring to precipitate the polysaccharide, and kept at 4 °C overnight. Finally, the polysaccharide precipitation was obtained by centrifugation for 10 min at 6000 rpm and washed sequentially with minimal amounts of ethanol, acetone and ether. Residue was dried to obtain the *Polyporus umbellatus* polysaccharide (PPS).

### 3.3. Optimization of Hydrolysis Conditions

Orthogonal experiment is a common and simple method to study multiple factors and multiple levels, which can optimize the process by analyzing the typical experimental results [22,29,51,52,53]. In order to investigate optimal hydrolysis conditions of PPS, the orthogonal L_9_ (3)^4^ experiments were designed. The influencing factors were considered, including hydrolysis time (factor A), hydrolysis temperature (factor B), and TFA concentration (factor C). In addition, each factor had three levels to be optimized. Nine hydrolysis experiments were conducted in sequence at different temperatures (100, 110, and 120 °C), hydrolysis times (3, 6, and 9 h), and TFA concentrations (2, 3, and 4 mol/L). The experimental conditions for the hydrolysis of PPS were listed in Appendix A.

### 3.4. Hydrolysis of Polysaccharide

Polysaccharide sample (20 mg) was infiltrated with 2 mL TFA [54] in a 10 mL ampoule that was sealed under a nitrogen atmosphere, and hydrolysis of the polysaccharide was performed. After cooling to room temperature, the resulting reaction solutions were centrifuged at 6000 rpm for 10 min, then the supernatant was evaporated to dryness under reduced pressure. The dried sample was for the following experiments.

### 3.5. Derivatization of Hydrolyzed Polysaccharide

The PMP derivatization of monosaccharides was carried out according to previous method [27,55,56,57] with minor modifications. The resulting monosaccharides were dissolved in 5 mL ammonia (28%). Then, 100 μL of solution were mixed with 0.5 mol/L methanolic solution of PMP (100 μL) in a 3-mL ampoule that was sealed under a nitrogen atmosphere. The mixture was allowed to react for 30 min at 70 °C in water bath, then cooled to room temperature and was concentrated to dryness under nitrogen. Water and chloroform (1.0 mL each) were added and mixed, and the organic phase was carefully removed. The extraction process was repeated three times and the aqueous layer was filtered through a 0.45-μm membrane prior to LC/MS analysis.

### 3.6. HPLC–MS Conditions

#### 3.6.1. HPLC Conditions

The chromatographic analysis was performed on a Shimadzu analytical HPLC system (Kyoto, Japan), consisting of DGU-20A3 degasser, two LC-20AD pumps, an SIL-20AC auto injector, a CTO-20A column oven, and an SPD-M20A detector. Sample separation was carried on a Shimpack VP-ODS C_18_ column (250 mm × 4.6 mm, 5 μm, Shimadzu, Kyoto, Japan) kept at 30 °C. The mobile phase was composed of 0.1 mol/L ammonium acetate (adjusted to pH 4.5 with acetic acid): acetonitrile (83:17, v/v). The flow rate was 1.0 mL/min. The injection volume was 20 μL. The UV spectra were recorded in the range of 200–400 nm, and the DAD was set at 245 nm.

#### 3.6.2. ESI–IT–TOF–MS Conditions

The HPLC system was coupled with ion-trap time-of flight (IT–TOF) mass spectrometer (Shimadzu LCMS–IT–TOF, Kyoto, Japan) via an ESI interface. The HPLC eluent at a flow rate of 0.2 mL/min was introduced into the ESI source after DAD detection. The MS scan range in positive mode was *m/z* 100–700. The temperatures of curved desolvation line (CDL) and block heater were both maintained at 200 °C. The capillary voltage, CDL voltage, and detector voltage were set at 4.5 kV, 10 V, and 1.75 kV, respectively. Nitrogen was used as the nebulizer gas at a flow rate of 1.5 L/min. The collision energy was adjusted to 70% in the HPLC–MS analysis, and the isolation width of precursor ions was 3.0 U. The data was acquired and processed by LC/MS solution software (version 3, Shimadzu, Kyoto, Japan) with a chemical formula predictor.

## 4. Conclusions

In this study, a simple, robust, and effective HPLC–DAD–ESI–IT–TOF–MS method with PMP derivatization was established for the analysis of monosaccharide composition of polysaccharide from *Polyporus umbellatus*. All 12 PMP-labeled derivatives of monosaccharides displayed high chemical stability, characteristic MS fragmentations for structural identification. Among them, ribose, lyxose, erythrose, and galacturonic acid were detected for the first time in *Polyporus umbellatus*. In addition, the optimized hydrolysis conditions of PPS by the orthogonal L_9_ (3)^4^ experiments will greatly facilitate further development of the medicinal polysaccharide from *Polyporus umbellatus*. This study provides an analytical approach and chemical basis for the quality control of *Polyporus umbellatus*.

## Figures and Tables

**Figure 1 molecules-24-02526-f001:**
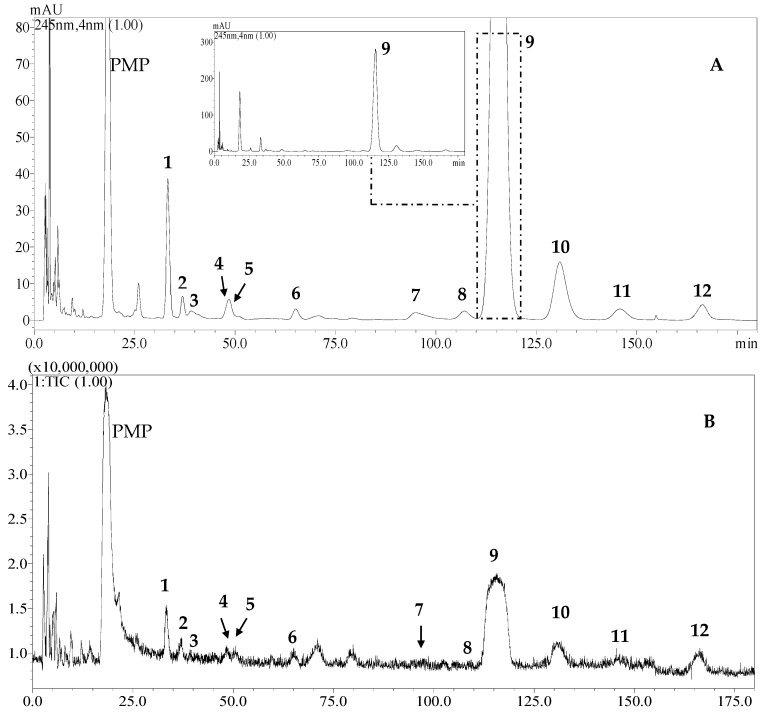
The HPLC chromatogram (**A**) and total ion chromatogram (TIC) in positive mode (**B**) of 12 1-phenyl-3-methyl-5-pyrazolone (PMP)-labeled monosaccharides. **1**, mannose; **2**, glucosamine; **3**, lyxose; **4**, rhamnose; **5**, ribose; **6**, erythrose; **7**, glucuronic acid; **8**, galacturonic acid; **9**, glucose; **10**, galactose; **11**, xylose; **12**, fucose.

**Figure 2 molecules-24-02526-f002:**
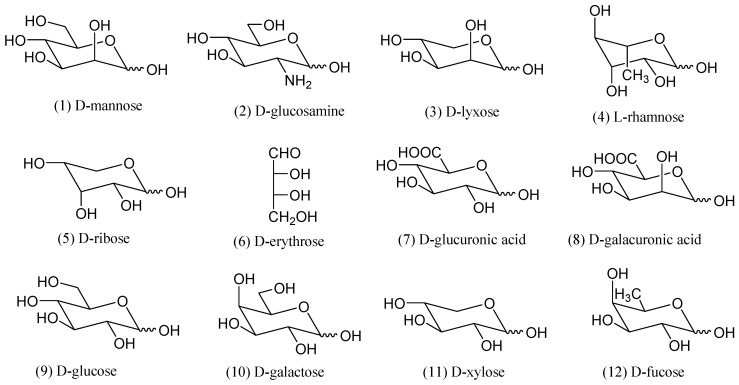
Chemical structures of 12 monosaccharides identified from *Polyporus umbellatus* polysaccharide (PPS).

**Table 1 molecules-24-02526-t001:** Characteristic ions of MS/MS for PMP-labeled monosaccharides (*m/z*).

t*_R_* (min)	Compound	[M + H]^+^ (Error in ppm)	Characteristic MS^2^ Fragments Ions of: [M + H]^+^											
			[M + H − H_2_O]^+^	C3-C4 Cleavage and −H_2_O	C2-C3 Cleavage and −H_2_O	[M + H − PMP]^+^	[M + H − PMP − H_2_O]^+^	[M + H − PMP − 2H_2_O]^+^	[M + H – PMP − 3H_2_O]^+^	C5-C6 Cleavage and − 2H_2_O	C4-C5 Cleavage and − 2H_2_O	C2-C3 Cleavage	C1-C2 Cleavage	[PMP + H]^+^
33.830	Mannose (1)	511.2201 (+2.74)	493.2283	403.1312	373.1751	337.1331	319.1064	301.1157	283.1149	271.1069	241.1037	217.1045	187.086	175.0904
36.548	Glucosamine (2)	510.2350 (+0.59)	492.2207	402.1679	372.1868	336.1679	319.1589^a)^	301.0892^b)^	283.1082^c)^	271.1083^d)^	241.0959^e)^	216.1313	187.0923^f)^	175.0904
40.298	Lyxose (3)	481.2061 (−4.36)	463.2209	-	373.1593	307.1321	289.1476	271.1193	253.1302	-	241.1071	217.1093	187.0893	175.0873
47.701	Rhamnose (4)	495.2243 (+1.01)	-	-	373.1729	321.1649	303.1136	285.1227	267.1122	-	241.0933	217.0929	187.0924	175.0902
48.832	Ribose (5)	481.2073 (−1.87)	463.2216	-	373.1750	307.1325	289.1153	271.1021	253.1001	-	241.0979	217.1053	187.0901	175.0919
64.678	Erythrose (6)	451.1977 (+0.22)	433.1797	-	373.1749	277.1583	259.1082	241.0970	-	-	-	217.0938	187.0943	175.0861
94.865	Glucuronic acid (7)	525.1972 (−1.52)	507.1776	-	373.1567	-	-	-	297.0914	271.0983	241.0941	217.1039	187.0796	175.0838
107.503	Galacuronic acid (8)	525.2005 (+4.76)	507.1841	-	373.1733	-	-	-	297.0766	271.0983	241.1252	217.1018	187.0809	175.0875
115.318	Glucose (9)	511.2187 (+0)	493.1923	403.1419	373.1652	337.1408	319.1341	301.1345	283.1135	271.1116	241.0986	217.0986	187.0884	175.0860
130.080	Galactose (10)	511.2187 (+0)	493.2219	403.1561	373.1693	337.1036	319.1108	301.1322	283.1110	271.1120	241.1001	217.0951	187.0804	175.0872
145.282	Xylose (11)	481.2072 (−2.08)	463.1764	-	373.1680	-	289.1448	271.1019	253.0949	-	241.1188	217.0712	187.0896	175.1019
166.837	Fucose (12)	495.2259 (+4.24)	477.2073	403.1800	373.1626	321.1555	-	285.1205	267.1102	-	241.1030	217.1096	187.0925	175.0906

t*_R_*: retention time; –: No detected; M: represent molecular weight of PMP-labeled monosaccharides. ^a)^ [M + H – PMP − NH_3_]^+^. ^b)^ [M + H – PMP − NH_3_ − H_2_O]^+^. ^c)^ [M + H – PMP − NH_3_ − 2H_2_O]^+^. ^d)^ [M + H – PMP − CH_2_O − NH_3_ − H_2_O]^+^. ^e)^ [M + H – PMP − 2CH_2_O − NH_3_ − H_2_O]^+^. ^f)^ [M + H – PMP − 4CH_2_O − CHNH_2_]^+^.

**Table 2 molecules-24-02526-t002:** Calibration curves, linear ranges, limits of detection (LODs), and limit of quantification (LOQs) of 10 monosaccharide derivatives.

No.	Analyte	Calibration Curve	R^2^	Linear Range (μmol/L)	LOD (μmol/L)	LOQ (μmol/L)
**1**	Mannose	y = 3.0804 × 10^4^x + 6.3221 × 10^4^	0.9998	20.20–202.04	0.19	0.63
**2**	Glucosamine	y = 7.2843 × 10^3^x − 7.3839 × 10^3^	0.9999	8.40–151.26	0.83	2.76
**3**	Lyxose	y = 2.1137 × 10^4^x + 1.9181 × 10^3^	0.9999	4.26–42.63	0.22	0.73
**6**	Erythrose	y = 5.3561 × 10^3^x +1.3761 × 10^4^	0.9998	12.14–133.54	0.54	1.80
**7**	Glucuronic acid	y = 2.5459 × 10^4^x − 4.0005 × 10^4^	0.9995	8.14–81.38	1.15	3.83
**8**	Galacuronic acid	y = 3.1860 × 10^4^x + 1.2310 × 10^4^	0.9999	3.96–39.61	0.37	1.23
**9**	Glucose	y = 2.3516 × 10^4^x + 1.5884 × 10^5^	0.9999	199.82–3596.80	0.29	0.97
**10**	Galactose	y = 2.7091 × 10^4^x + 7.5494 × 10^3^	0.9998	41.33–413.33	0.74	2.47
**11**	Xylose	y = 1.7797 × 10^4^x + 5.0060 × 10^4^	0.9996	16.95–152.51	0.57	1.90
**12**	Fucose	y = 1.6309 × 10^4^x + 5.7003 × 10^4^	0.9991	8.07–80.67	0.78	2.60

**Table 3 molecules-24-02526-t003:** Precision, repeatability and stability of 10 monosaccharide derivatives. RSD: relative standard deviation.

No.	Analyte	RSD (%)		
		Precision	Repeatability	Stability
**1**	Mannose	0.39	0.49	0.67
**2**	Glucosamine	1.03	1.09	1.23
**3**	Lyxose	1.13	1.22	1.30
**6**	Erythrose	0.62	0.69	0.64
**7**	Glucuronic acid	1.14	1.13	1.05
**8**	Galacuronic acid	0.53	0.71	0.75
**9**	Glucose	0.48	0.70	0.73
**10**	Galactose	0.94	1.40	1.02
**11**	Xylose	1.10	1.22	1.15
**12**	Fucose	0.84	0.88	1.10

**Table 4 molecules-24-02526-t004:** Recovery analysis of 10 monosaccharides of PPS (*n* = 5).

No.	Analyte	Content (nmol/mg)	Spiked (nmol/mg)	Mean Found (nmol/mg)	Recovery (%)	RSD (%)
**1**	Mannose	234.88	230	470.88	102.61	1.79
**2**	Glucosamine	111.70	110	217.41	96.10	2.01
**3**	Lyxose	291.81	291	576.87	97.96	1.25
**6**	Erythrose	271.27	271	545.06	101.03	2.36
**7**	Glucuronic acid	72.81	72	143.73	98.50	1.56
**8**	Galacuronic acid	46.14	46	90.90	97.30	1.37
**9**	Glucose	3079.52	3000	6091.52	100.40	0.96
**10**	Galactose	484.26	485	977.36	101.67	1.01
**11**	Xylose	220.69	220	439.59	99.50	1.16
**12**	Fucose	289.87	290	590.60	103.70	1.43

**Table 5 molecules-24-02526-t005:** Determination results of the monosaccharides in PPS from 12 samples of *Polyporus umbellatus*.

Sample	Origin	Contents (nmol/mg, *n* = 3)									
		Mannose	Glucosamine	Lyxose	Erythrose	Glucuronic Acid	Galacuronic Acid	Glucose	Galactose	Xylose	Fucose
S1	Shaanxi	234.88	111.70	291.81	271.27	72.81	46.14	3079.52	484.26	220.69	289.87
S2	Shanxi	145.22	61.86	302.68	460.78	37.15	29.94	2769.76	351.48	342.07	363.18
S3	Shanxi	107.39	99.07	217.79	331.03	31.02	27.53	1473.43	239.13	282.08	289.29
S4	Yunnan	153.97	106.59	182.08	520.85	52.42	55.43	3279.84	268.36	479.31	390.09
S5	Yunnan	175.84	102.97	221.64	439.15	47.16	38.33	2761.47	437.67	306.93	403.57
S6	Shaanxi	136.04	80.02	94.31	373.67	77.96	80.28	3329.49	516.64	284.02	291.84
S7	Shaanxi	160.08	75.85	129.04	355.81	41.13	35.81	2814.63	444.62	289.06	296.59
S8	Henan	82.81	63.42	115.88	153.51	26.26	16.21	1222.47	266.39	101.85	169.37
S9	Hebei	55.98	69.50	77.42	157.88	20.88	15.67	1284.87	126.54	110.60	117.21
S10	Sichuan	69.01	60.01	136.19	150.25	23.89	17.32	1379.81	253.40	129.34	169.23
S11	Jilin	76.03	64.66	50.73	146.89	34.02	32.02	1094.02	222.16	117.18	130.78
S12	Gansu	174.9	153.11	68.71	321.26	70.45	70.53	3474.96	308.13	261.42	224.71

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
