# Peer review of "Quantitative Analysis of Polysaccharide Composition in Polyporus umbellatus by HPLC–ESI–TOF–MS"

_molecules, 2019, doi:10.3390/molecules24142526_

Round 1

Reviewer 1 Report

The manuscript describes the application of an LC-MS method with PMP derivatization for the monosaccharide compositions analysis of Polyporus umbellatus, a medicinal fungus used in Asia.

The authors analysed 12 samples of Polyporus umbellatus from different origins.  The monosaccharide composition analysis has been carried out through polysaccharide hydrolysis, monosaccharides derivatization with PMP and then identified and quantified by HPLC-MS/MS analyses. The results are properly described. Please, check some minor typing mistakes throughout the text.

Reviewer 2 Report

The authors presented a method for determining the polysaccharide composition based on the monosaccharide profile determined by the HPLC-ESI-TOF-MS method. The manuscript is concise and written clearly. The number of tables and drawings is adequate. The described methods and presentation of results may be interesting for a wide group of readers dealing with polysaccharide isolation and the use of HPLC-MS methods. I would recommend the presented manuscript for puboication in its current form.
